# Transformation of Amorphous Terbium Metal–Organic Framework on Terbium Oxide TbO_x_(111) Thin Film on Pt(111) Substrate: Structure of Tb_x_O_y_ Film

**DOI:** 10.3390/nano12162817

**Published:** 2022-08-17

**Authors:** Helena Brunckova, Erika Mudra, Magdalena Streckova, Lubomir Medvecky, Tibor Sopcak, Ivan Shepa, Alexandra Kovalcikova, Maksym Lisnichuk, Hristo Kolev

**Affiliations:** 1Institute of Materials Research, Slovak Academy of Sciences, Watsonova 47, 040 01 Kosice, Slovakia; 2Institute of Catalysis, Bulgarian Academy of Sciences, Acad. G. Bonchev St., 1113 Sofia, Bulgaria

**Keywords:** metal–organic frameworks, solvothermal synthesis, amorphous TbMOF, thin film, terbium oxide, microstructure

## Abstract

The present study is focused on the synthesis and structural properties of amorphous terbium metal–organic framework thin film (TbMOF-TF) and its transformation to terbium oxide by pyrolysis at 450 °C in the air. The crystalline (cTbMOF) and amorphous (aTbMOF) films were prepared by solvothermal synthesis using different amounts (0.4 and 0.7 mmol) of the modulator (sodium acetate), respectively. The powders were characterized by differential scanning calorimetry (DSC), thermogravimetry (TG), Fourier transform infrared (FTIR), Raman spectroscopy, and scanning electron microscopy (SEM). The varied chemical composition of the surface of TbMOFs and Tb_x_O_y_ was investigated by X-ray photoelectron spectroscopy (XPS). X-ray diffraction (XRD) and transmission electron microscopy (TEM) revealed that aTbMOF had been fully transformed to a Tb_4_O_7_ phase with a cubic crystal structure at 450 °C. The amorphous aTbMOF-TF film was prepared by dropping a colloidal solution of amorphous precursor nanocrystals on the SiO_2_/Si substrates covered with Pt as an interlayer. XPS confirmed the presence of Tb in two states, Tb^3+^ and Tb^4+^. The amorphous film has a rough, porous microstructure and is composed of large clusters of worm-like particles, while terbium oxide film consists of fine crystallites of cubic fluorite cF-TbO_x_, c-Tb_4_O_7,_ and c-Tb_2_O_3_ phases. The surface topography was investigated by a combination of confocal (CM) and atomic force microscopy (AFM). The amorphous film is porous and rough, which is contrast to the crystalline terbium oxide film.

## 1. Introduction

Metal–organic frameworks (MOFs), which belong to the subclass of porous coordination polymers (PCPs), are materials built from metal ions and organic bridging ligands [1,2]. MOFs have attracted much attention in various fields where they are used, such as sensor technology, catalysis, drug delivery, thin films, luminescence, gas storage, and separation [1].

Amorphous materials are excellent due to their “dangling bonds” and, in many applications, are more active than their crystalline counterparts [3,4,5,6]. While most research efforts have focused on crystalline MOFs, only a few amorphous (aMOFs) have been reported. From the study of Bennett et al. [7], it can be concluded that aMOFs created by introducing pressure, temperature, or ball milling still preserve the basic building blocks and short-range connectivity. FeMOF has a nanocomposite structure containing both crystalline and amorphous phases [8]. Lanthanide metal–organic frameworks (LnMOFs) are currently gaining increasing attention due to their potential applications in areas such as gas adsorption, catalysis, magnetism, and especially for their photoluminescence properties [9,10].

Nowadays, MOFs are available in several structures, such as nanocrystals (NCs), nanospheres, nanosheets, needles, hierarchical monoliths, thin films (TFs), membranes, and glasses [11]. Amongst these structures, MOF-TFs have attracted more attention due to their great potential in the development of nanotechnology applications such as optics, photonics, electronics, catalytic coatings, solar cells, batteries, and supercapacitors [11]. MOF-TFs deposited on substrates of variant functions have various applications [11,12]. In terms of fabrication techniques, commonly used procedures are known, such as hydro/solvothermal synthesis, layer-by-layer (LBL), and electrochemical methods. The coordination modulation method was first described by Kitagawa and Ferey for the synthesis of MOF nanocrystals [13,14]. Cai et al. and Li et al. later adopted this method to synthesize LnMOF nanocrystals using some monocarboxylic acids and carboxylate salts as modulators [10,15]. In special conditions, when redundant amounts of the modulator are added, the synthesis can lead to the origin of amorphous aMOF-TF [16]. Crystalline TbMOF as Tb(BTC)(H_2_O), prepared by the solvothermal synthesis, with the addition of 1,3,5-benzene tricarboxylic acid (BTC) and sodium acetate (NaOAc), changed when the amount of NaOAc was increased up to 0.7 mmol, resulting in the formation of the product in the amorphous phase [10].

Terbium oxides are used in various scientific and technological applications, such as optical, ceramic, and chemical ones [17]. The conventional thermolysis of Tb salts in the air usually yields oxygen-deficient fluorite terbium oxides TbO_x_ of type x ≈ 1.75 [17]. These oxides were—by mistake—considered to be single-phase intermediate Tb_4_O_7_, but in fact, such a crystalline phase cannot be obtained, and the oxides TbO_1.75_ appeared to be two-phase mixtures of the related fluorite trigonal Tb_7_O_12_ and triclinic Tb_11_O_20_ [17,18]. Porous bimetallic Ce_1−x_Tb_x_O_y_ crystals in powder form were prepared by the direct pyrolysis of Ce_1−x_Tb_x_-MOFs precursors [19]. It is known that in binary Tb oxides, the cations can exist in two states: Tb^3+^ and Tb^4+^ [17]. These distinct Tb^3+^/Tb^4+^ cation equilibria lead to different phase transitions, such as Tb0_2_, Tb_11_O_20_, Tb_4_O_7_, and Tb_7_O_12_ [20]. Terbium oxides include the stoichiometric oxides Tb_2_O_3_ and TbO_2_, as well as non-stoichiometric oxides TbO_1.714_, TbO_1.750_, TbO_1.818_, TbO_1.833,_ and TbO_1.875_ [21]. Non-stoichiometric oxides are transition compounds of hybrid efficiency with compositions ranging between Tb_2_O_3_ and TbO_2_. The highest terbium oxides (Tb_11_O_20_ and Tb_7_O_12_) are unstable and are subject to decomposition at high temperatures with the removal of oxygen. Tb_11_O_20_ converts to Tb_7_0_12_ at ~600 °C in air and Tb_7_O_12_ to Tb_2_O_3_ above 1000 °C [21].

In films, all lanthanides can create stable sesquioxides (Ln_2_O_3_); only Pr and Tb are known to can form a wide range of intermediate oxides of Ln^3+^ and Ln^4+^ states (e.g., Pr_6_O_11_, Tb_4_O_7_) [22]. Terbium oxide films possess a wide variety of functional properties and can be used as gas sensors, luminophores, and optical electronics [23,24]. Polycrystalline terbium oxide films were grown on Si(111) substrates by Tb(dpm)_3_ vapor decomposition [23,24]. After annealing in air at temperatures of 400 and 650 °C, the films of Tb_11_O_20_, Tb_7_O_12,_ and Tb_4_O_7_ differed in composition [23,24]. Lee et al. [25,26] prepared terbium oxide films with varying thicknesses by reactive physical vapor deposition (RPVD) of metallic terbium on Pt(111) substrate. Hexagonal film growth proceeds according to the Stranski–Krastanov mechanism, in which oxygen-deficient cubic fluorite cF-TbO_x_(111) initially forms a well-connected wetting layer, and the phases of δ-Tb_11_O_20_, ι-Tb_7_O_12_, TbO_x_ (x = 1.75), and cF-TbO_3_ are progressively formed [26]. Similarly, terbium oxide films prepared by RPVD deposition on Cu(111) contained crystalline hexagonal dendrites of cF-TbO_x_(111) [22].

In this work, we investigated the structural and morphological properties of amorphous TbMOF (powder and film) and its thermal decomposition to terbium oxide during pyrolysis at 450 °C. The TbMOF-TF thin film was prepared by solvothermal synthesis with a [Tb(btc)] structure (BTC: Benzene-1,3,5-tricarboxylate) on a Pt/SiO_2_/Si substrate. The chemical composition, microstructure, and topography of the TbMOF-TF and Tb_x_O_y_-TF films were analyzed by XRD, FTIR, Raman spectra, SEM, TEM, AFM, and CM methods. We reported a facile strategy to prepare amorphous aMOFs by the introduction of Tb and further pyrolysis at 450 °C to form a unique terbium oxide thin film with a thickness of ~400 nm on a Pt/SiO_2_/Si substrate. The novelty is a fabricated porous film, which is composed of morphologically different phases TbO_x_, Tb_4_O_7,_ and Tb_2_O_3_ nanoparticles. The prepared films are suitable candidates for optoelectronic applications as gas sensors.

## 2. Materials and Methods

### 2.1. Chemicals and Materials Used

All the chemicals used (including solvents) were of analytical grade, purchased from Sigma-Aldrich, and used as-is without any further purification. 1,3,5-benzenetricarboxylic acid (BTC), Tb(NO_3_)_3_·6H_2_O, *N*,*N*-dimethylformamide (DMF), sodium acetate (NaOAc), ethanol (EtOH), and deionized water were used for the solvothermal synthesis of the two MOF types, both crystalline (cTbMOF) and amorphous (aTbMOF).

### 2.2. Preparation of cTbMOF and aTbMOF

The TbMOFs were prepared through modified solvothermal synthesis [7,10,19] according to previous work [27,28], as described in Figure 1. Terbium(III) nitrate hydrate Ln(NO_3_)_3_·6H_2_O (1.0 mmol, 0.45 g) and H_3_BTC (1.0 mmol, 0.21 g) were dissolved in 30 mL of the DMF/H_2_O (1:1 *v/v*) solvents mixture together with the NaOAc modulator. The preparation procedures for cTbMOF and aTbMOF were the same and were performed using different amounts of modulator NaOAc for crystalline (0.4 mmol, 0.033 g) and amorphous (0.7 mmol, 0.055 g) TbMOFs. The two solutions of cTbMOF and aTbMOF were mixed at 25 °C for 1 h and then heated up to 60 °C, held for 48 h, and cooled down to room temperature. After synthesis, the products were isolated by centrifugation and washed several times with ethanol and water, respectively, and then dried in air. The prepared cTbMOF and aTbMOF had a yield of 70% (0.339 g) and 72% (0.349 g), respectively, without elemental analysis. The Tb_x_O_y_ crystals were prepared by calcination of the as-synthesized aTbMOF at 450 °C in the air for 2 h. The framework of the Ce_1−x_Tb_x_MOF crystals was prepared using a similar method and started to collapse at 350 °C [19].

### 2.3. Preparation of cTbMOF-TF and aTbMOF-TF Thin Films

With the aid of crystalline and amorphous TbBTC colloidal seeds, the cTbMOF-TF and aTbMOF-TF thin films were prepared based on the substrates (Pt/SiO_2_/Si) at conditions similar to those described in Figure 1. The powders of cTbMOF and aTbMOF were individually redispersed in H_2_O, so milky colloidal solutions were obtained with the concentrations of TbBTC of 0.03 g/mL. The colloidal solutions of the TbMOF were deposited onto the substrates by drop-casting. The mixed slurries were deposited on pre-cleaned silicon substrates covered with SiO_2_ and Pt. As the initial substrates, p-type silicon [100] single-crystal wafers with a diameter of 50 mm and 270 μm were used. The Pt layer was applied to the substrate by ion-sputtering from a Pt metal target. The final thickness of the Pt layer obtained on the surface of the 250 nm SiO_2_ film was about 20 nm. Continuous thin films of TbMOF were directly grown on Pt(111) [25,26]. MOFs were grown by repeating two drop-casting deposition cycles. The thickness of the MOF films can be easily controlled by the TbBTC concentration in the colloidal solution. The deposited films were then dried at 60 °C for 2 h. The prepared crystalline and amorphous TbBTC films were denoted as cTbMOF-TF and aTbMOF-TF. The Tb_x_O_y_-TF thin film was prepared by calcination of the aTbMOF-TF thin film at 450 °C in the air for 2 h.

### 2.4. Characterization of the Obtained Powders (TbMOF and Tb_x_O_y_) and Thin Films (TbMOF-TF and Tb_x_O_y_-TF)

The thermal properties and the decomposition process of TbMOFs were analyzed by differential scanning calorimetry (DSC) coupled with thermogravimetric (TG) analysis (JUPITER STA 449-F1, NETZSCH, Selb, Germany) in platinum crucibles, temperature range of 50 to 1000 °C, and 10 °C min^−1^ heating rate in air. The chemical/phase composition of the obtained samples was studied by FTIR (Shimadzu (Kyoto, Japan) IRAffinity 1 with KBr pellets) and Raman spectroscopy (Raman spectroscope HORIBA BX 41TF, Kyoto, Japan); the phase composition was determined by X-ray diffraction analysis (XRD, X^’^ Pert Pro, Philips, Amsterdam, Netherlands, with CuK_α_ radiation). The diffraction patterns were recorded in a 2θ range from 10 to 60° with a 4°min^−1^ scan rate. The additional composition and valence state inquiry was assessed by X-ray photoelectron spectroscopy (XPS). XPS measurements were carried out on an ESCALAB MkII (Thermo Fisher Scientific, Waltham, MA, USA) electron spectrometer equipped with a twin-anode MgK_α_/AlK_α_ non-monochromated X-ray source. The measurements were taken with an AlK_α_ X-ray source (1486.6 eV). The energy range was scaled using a standardizing C 1s line of acquired hydrocarbons to 285.0 eV for the electrostatic sample charging. The obtained data were analyzed and processed by SpecsLab2 CasaXPS 2.3.25 software (Casa Software Ltd., Berlin, Germany). The processing of the recorded spectra included a subtraction of X-ray satellites and Shirley-type background. The relative concentrations of the different chemical species were determined based on the normalization of the peak areas to their photoionization cross-sections, calculated by Scofield.

The surface morphologies of the samples were examined using Scanning Electron Microscopy (FESEM/FIB, Auriga Compact, Carl Zeiss, Oberkochen, Germany) equipped with an energy-dispersive X-ray analyzer (EDS) and high-resolution transmission electron microscopy (TEM, JEOL-JEM 2100F, Tokyo, Japan), using a scanning transmission electron microscopy mode (STEM) and EDS (Oxford Energy TEM250, Abingdon, UK). Before scanning, all samples were coated with carbon to enhance their conductivity. The surface topography and roughness of the films were estimated by a combination of confocal (CM, PluNeox 3D optical surface profiler by SENSOFAR (Barcelona, Spain) with 20× objective) and atomic force microscopy (AFM, Dimension ICON, by Veeco Instruments, Plainview, NY, USA).

## 3. Results and Discussion

### 3.1. Structural Characterization of Tb-Based Powders and Thin Films

DSC and TG curves of crystalline cTbMOF, amorphous aTbMOF, and Tb_x_O_y_ powders are shown in Appendix A. Both samples of MOF have similar curves up to 400 °C. The weight losses in the temperature ranges of 150–220 °C and 220–350 °C are cognates with the release of trapped DMF solvent and water [29]. The notable difference between crystalline and amorphous MOF appears in the range of 410–650 °C. In this temperature region, the weight loss originates from the decomposition of the organic linkers of the framework. Amorphous MOF decomposes slowly by forming CO_2_ with an onset at ~380 °C and ending at ~650 °C, whereas crystalline start to release CO_2_ at higher temperatures of ~400 °C and up to ~670 °C. The TbMOFs have great thermal stability. After complete dehydration of terbium oxide (Tb_x_O_y_ sample) in the temperature range of ~50–200 °C, the carbonate structure of Tb_2_(CO_3_)_3_ was formed, which was thermally stable until 400 °C, after which it completely decomposed into terbium oxide Tb_4_O_7_ (wide exo peak at 700 °C) in the temperature range between 450 and 760 °C [30,31]. A small endo peak at 840 °C represents the transformation of Tb_4_O_7_ to Tb_2_O_3_ oxide [18]. Based on the DSC and TG results, it was confirmed that the amorphous aTbMOF was thermally decomposed into terbium oxide at 450 °C.

Figure 1 shows the FTIR and Raman spectra of amorphous (aTbMOF), crystalline (cTbMOF), and aTbMOF pyrolyzed at 450 °C (sample Tb_x_O_y_) powders and corresponding films (TF). In the FTIR spectra (Figure 1a), the wide peak at 3435 cm^−1^ is assigned to υ (O–H) groups. The effect of acetate groups from sodium acetate for cTbMOF and aTbMOF can be noticed in the regions at 2998, 2780, and 2430 cm^−1^, which are assigned to stretching υ(C–H) vibrations. The peak at 1630 cm^−1^ belongs to ν(C=O) of DMF. In the spectra, the bands in zones 1560–1518 cm^−1^ and 1385 cm^−1^ were marked as stretching vibrations of the COO^−^ groups υ_as_ and υ_s_, respectively. The powerful peaks provide the C–H bending benzene vibrations that shifted to the region of 765 and 720 cm^−1^ [28]. The peak that appeared at 565 cm^−1^ can be assigned to the stretching vibration of Tb–O. The structural designation of the TbMOFs is marked as [Tb(btc)(H_2_O)(dmf)] [27,28]. The peaks in the 1630–1370 cm^−1^ region disappear in the Tb_x_O_y_ sample, indicating that the TbMOF skeleton structure collapsed [19]. From FTIR analysis, we can see that crystalline Tb_x_O_y_ is a carbonate structure of Tb_2_(CO_3_)_3_·nH_2_O [32]. The bands located at 1440, 1380, 1060, and 876 cm^−1^ can be assigned to the carbonate (CO_3_)^2−^ structure [21,32], whereas the bands at 3440 and 1615 cm^−1^ are assigned to the ν(O–H) and ν(HOH) modes of vibration of crystalline water, respectively [21].

The Raman spectra of powders and films are depicted in Figure 1b. The spectra of crystalline and amorphous samples of the same chemical composition can be significantly different, primarily because of the presence or absence of spatial order. Exceptional peaks at about 1795; 1950; and in the region of 2350–2800 cm^−1^, corresponding to D, G, and 2D bands, respectively, are also visible on the spectra. The peaks of the powders are sharper than those of the films. The development of the cTbMOF was confirmed by the detection of Raman bands at 1525, 1895, and 2540 cm^−1^, which were identified as ν(COO^−^), ν(C=O), and ν(C–H), respectively, according to the literature [3]. The lines in the region of 2800–3100 cm^−1^ are ascribed to the stretching of the –CH group and asymmetric stretching of -NH bonds. Raman spectra of the films have broadened peaks. The peak at 520 cm^−1^ corresponds to the characteristic band of Tb–O [3,33]. Terbium oxide Tb_4_O_7_ has been reported as a mixed-valence compound with a non-stoichiometric structure [34]. The Tb_x_O_y_ powder shows one broadened peak at 620 cm^−1^. In the Tb_x_O_y_-TF film, both the intensities of D and G peaks increase [23]. Raman spectra show that the carbonization and Tb_x_O_y_-TF formation temperature is >400 °C. In agreement with the XRD, the FTIR spectra (Figure 1a and Figure 2a) for annealed aTbMOF sample also represent the remaining unchanged main building block of the MOF structure. The band recorded at 530 cm^−1^ assigned to the Tb–O bond [35] and sharp bands at 2485, 2845 cm^−1,^ and 3523 cm^−1^ are (C–H) vibrations. In the FTIR and Raman spectra results, the bands consequent from the presence of ligands in aTbMOF were deleted, which is a qualitative confirmation of terbium oxide formation from a carbonate structure.

The crystalline cTbMOF complex is 3D open framework, and each asymmetric unit contains one eight-coordinated Tb^3+^ ion, one BTC ligand (C_9_H_3_O_6_), two coordinated DMF molecules (C_3_H_7_NO), and one free guest water molecule (H_2_O) [36,37] as Tb(BTC)(DMF)_2_·H_2_O. In Appendix A, each Tb^3+^ ion is coordinated with eight oxygen atoms from four BTC ligands through two chelating bidentate carboxylate groups, two monodentate carboxylate groups, and two terminal DMF molecules. The empirical formula is C_15_H_19_N_2_O_9_Tb [36].

Figure 2 shows the XRD patterns for both samples (crystalline and amorphous): TbMOF powders as-synthesized and after thermal treatment at 450 °C (Tb_x_O_y_) of the same film. For cTbMOF, the results were compared with the crystallographic data in the Cambridge Database: CIF no. 617492 for Tb(BTC)(DMF)_2_·H_2_O; the match confirms the expected tetragonal phase for TbBTC [38]. As displayed in Figure 2a, all peaks of the sample synthesized with 0.4 mmol sodium acetate agree with the CIF, indicating they are phase Tb(BTC)(DMF)_2_·(H_2_O). When the amount of sodium acetate is increased to 0.7 mmol, an amorphous phase occurs [10,29]. For this sample, there is only one broad “hump” caused by diffuse scattering. This is a typical characteristic of amorphous aMOFs [5,9]. In XRD of Tb_x_O_y_ powder (Figure 2a), the peaks at 450 °C at 29.52, 33.9°, 48.6°, and 57.8° 2θ correspond to the (111), (200), (202), and (311) planes of Tb_4_O_7_ (PDF no. 13-0387). This indicates that after pyrolysis, the aTbMOF and cTbMOF precursor have been fully transformed into Tb_4_O_7_ with a cubic crystal structure [19,39]. Figure 2b shows the XRD pattern for cTbMOF-TF, aTbMOF-TF, and Tb_x_O_y_-TF thin films on Pt/SiO_2_/Si substrates. Similar to the XRD of crystalline powder, the cTbMOF-TF film (TbBTC phase) reveals peaks at 10.5, 11.5, 16.1, 21.3, and 29.3° (2θ) and Pt and Si peaks from the substrate. Another film is the amorphous aTbMOF-TF, which pyrolyzed to (Tb_x_O_y_-TF) consisting of three phases, cubic fluorite cF-TbO_x_ (x = 1.75), PDF no. 03-065-7172) [40]; cubic c-Tb_4_O_7_ (PDF no. 13-0387) [21,23,41,42,43]; and cubic c-Tb_2_O_3_ (PDF no. 19-1326) [32]. The XRD spectrum of Tb_x_O_y_-TF suggests that the film has binary terbium oxides with Tb^4+^ and Tb^3+^ states. The pyrolysis of TbMOF or Tb salts in the air generally yields oxygen-deficient, fluorite-related terbium oxides TbO_x_ with a composition of x ≈ 1.75–1.83 [17]. Intermediate Tb_11_O_20_, Tb_24_O_44_, and Tb_48_O_88_ phases in the terbium oxide TbO_x_ system have a fluorite structure [40]. Two oxides with the composition of Tb_11_O_20_ and Tb_7_O_12_ are probably formed at a temperature below 400 °C [23]. The TbO_1.75_ oxide appeared to represent a two-phase mixture of the fluorite-related stable ι-Tb_11_O_20_ (triclinic structure) and metastable δ-Tb_7_O_12_ (rhombohedral structure) [17]. The Tb_7_O_12_ phase is transformed to c-Tb_4_O_7_ and Tb_2_O_3_ at a temperature above 400 °C with corresponding mixtures of +3 and +4 valences [20,44]. Based on the XRD results, characteristic peaks of terbium oxide correspond with the crystalline structure of c-Tb_4_O_7_ in the Tb_x_O_y_ precursor and three phases (cF-TbO_x_, c-Tb_4_O_7,_ and c-Tb_2_O_3_) in the Tb_x_O_y_-TF film, and we did not observe the characteristic peaks of aTbMOF.

### 3.2. XPS Characterization of the Surfaces of Powders and Films

The XPS survey spectra for aTbMOF powder and aTbMOF-TF and Tb_x_O_y_-TF films are shown in Appendix A, where all peaks corresponding to the characteristic electronic transitions of Tb, O, N and C and Pt and Si (from the substrate) for films are visible. The Tb_x_O_y_-TF film contains carbon impurities. Tb 3d peak overlaps with the Auger peak of CKLL, and Tb 4d overlaps with Si 2s.

XPS core level spectra of Tb 3d, O 1s and C 1s for three samples are observed in Appendix A. The highest peaks for Tb 3d and Tb 4d are in Tb_x_O_y_-TF film. The XPS Tb 3d spectrum for the samples represents two peaks at 1245 and 1277 eV, assigned to 3d_5/2_ and 3d_3/2_ of Tb^3+^, respectively. The presence of Tb^4+^ was found in a small peak at 1250 eV [28]. The XPS spectrum of O 1s and C 1s in all of the samples showed peaks centered at 533 and 285 eV, respectively. The total concentration in atomic percent of Tb, O, C and N elements on the surface of amorphous powder and both films was determined in Appendix A.

XPS core-level spectra of Tb 3d, O 1s, and C 1s for aTbMOF-TF and Tb_x_O_y_-TF films are depicted in Figure 3. The content of Tb, O, and C elements on the surface of the aTbMOF-TF film in at.% was determined as 0.56, 30.22, and 30.25 %, respectively. In Tb_x_O_y_-TF film, the Tb and O content increases to 9.87 and 44.34%, respectively, but the C content decreases to 27.29%. The high-resolution (HR) XPS spectrum of the aTbMOF-TF film shows two peaks centered at 1276.9 (3d_3/2_) and 1242.5 eV (3d_5/2_), similar to Tb_x_O_y_-TF film (1276.5 and 1241.8 eV).

Figure 3 shows the detailed Tb 3d core-level XPS spectra together with the deconvolution analysis of the various Tb valence states for both films. Since the Tb content in the MOF film is low (0.56 at. %), the intensity of both peaks (Tb 3d_5/2_ and Tb 3d_3/2_) is quite low, and the Tb 3d_5/2_ (1242.5 eV) peak overlaps with the Auger peak of CKLL; therefore, the analysis can only be realized over the Tb 3d_3/2_ peak. The Tb 3d_5/2_ band can be resolved into the two sub-bands peaking at 1274.6 (Tb^3+^) and 1276.9 eV (Tb^4+^). In the terbium oxide film (Tb 9.87 at.%), there are two principal Tb 3d_5/2_ and Tb 3d_3/2_ bands peaking at 1241.8 and 1276.5 eV, respectively. Two sub-bands peaking at 1238.9 and 1273.7 eV can be ascribed to the Tb^3+^ state, and the two peaking at 1241.8 and 1276.5 eV can be ascribed to the Tb^4+^ state. According to XPS analysis results, Zhu et al. [42] reported Tb_2_O_3_ (Tb^3+^) and TbO_2_ (Tb^4+^) in the Tb_4_O_7_ film.

The O 1s peak was deconvoluted, and three peaks centered at 531.8, 532.7, and 533.7 eV were attributed to the Tb–C–O, H–C–O, and O–H, bonds of aTbMOF-TF. The binding energy of O 1s changes from its initial 528.9 eV (Tb–O) via 530.7 (O–H) to 532.5 eV (H–C–O) in the Tb_x_O_y_-TF film. In the aTbMOF-TF film, the C 1s peak was deconvoluted into peaks centered at 285.0, 286.6, and 289.1 eV, representing C–H, C–O, C–O–H, and C=O/C–O–C, respectively. In the Tb_x_O_y_-TF film, the peak at 285.0 eV is related to C-H bonds. The peak at 288.7 eV is related to C-O bonds as carbonates (CO_3_)^2−^. The XPS C 1s, O 1s, and Tb 3d core level spectra of amorphous film and its annealed sample are supported well by the XRD, Raman, and FTIR results. Qualitatively appreciable change is detected for the sample annealed up to 450 °C, with clear framework decomposition (mainly in the form of CO_2_ loss) and carbonization [35].

Appendix A shows the Pt 4f spectra of the Pt/SiO_2_ substrate in aTbMOF-TF film. It is the Pt 4f (4f_7/2_ and 4f_5/2_) doublet signal from the spectrum at 71.1 and 74.5 eV, respectively. The binding energy (Pt 4f_7/2_) for the sample was 71.1 eV, which is the same as the corresponding value of 71.2 eV for bulk platinum metal (Pt^0^) [45]. After annealing of the film at 450 °C (Tb_x_O_y_-TF), the Pt 4f signal shifted by 0.2 eV to a lower BEs (Appendix A). Appendix A shows the XPS Si 2s and Tb 4d core-level spectra recorded in the high-resolution mode at 149.9 (Si^4+^) and 149.9 eV (Tb^3+^). The XPS spectrum of the Si 2p line at 103.2 eV (Si^4+^, SiO_2_) in the Tb_x_O_y_-TF sample is shown in Appendix A. In the Pt/SiO_2_/Si substrate, Pt 4f is about 0.5 eV higher BE in comparison to the samples with both films; therefore, we assume that we have some interaction between Pt and silicon substrate. A binding energy of 72.8 eV is typical for Pt/Si, Pt_2_Si, Pt(OH)_2_, and even for PtSi [45]. XPS confirmed the presence of Tb in two Tb^3+^ and Tb^4+^ valence states in both films.

### 3.3. Morphological Characterization of Powders

The surface morphology of the samples was characterized using SEM and TEM. The images of TbMOF (crystalline and amorphous) and Tb_x_O_y_ powders are shown in Figure 4 and Appendix A. Observation of crystalline cTbMOF in Figure 4a reveals that the nano straw sheaves ~500 nm in length contained smaller rice-like crystals. As shown in Figure 4b, nanoparticles of TbBTC with an average size of 200 nm were observed by TEM, similar to particles of another lanthanide DyBTC with a NaOAc modulator [29]. Amorphous aTbMOF (confirmed by XRD analysis) was obtained when the amount of NaOAc was 0.7 mmol (Figure 4c,d and Appendix A). The high-resolution transmission electron microscopy (HR-TEM) image reveals no lattice fringes, and the selected area electron diffraction (SAED) shows no perceptible diffraction rings (Figure 4d and Appendix A). Figure 4e and Appendix A depict the SEM surface morphology of the pyrolyzed aTbMOF sample. This Tb_x_O_y_ powder exhibits a sponge-like surface. In addition, the surface of samples is covered with randomly distributed pores [21]. During annealing, uniform nanoparticles of Tb oxide are formed and aggregated to Tb_4_O_7_ crystals (Ostwald ripening) [19]. The TEM images of the Tb_x_O_y_ powder (Figure 4f and Appendix A) show that the particles of Tb_4_O_7_ were irregular in shape. The aggregates are composed of small primary nanoparticles (10–20 nm). This range of sizes is in agreement with the size (6–12 nm) for Tb_4_O_7_ reported in the literature [21].

The SEM images and EDS mapping (Figure 5 and Figure 6) of aTbMOF and Tb_x_O_y_ samples further detected the presence of Tb, O, and C as incorporated elements in the amorphous sample and Tb_4_O_7_, respectively. EDS spectra of aTbMOF and Tb_x_O_y_ show significant peaks of elements in surface samples. The surface composition for aTbMOF in at.% of Tb, O, C, and N were determined as 5.1, 37.1, 50.5, and 7.3, respectively. The values calculated for the assumed formula of crystalline [Ln(btc)(dmf)_2_(H_2_O)], C_15_H_19_N_2_O_9_Tb is C_9.9_H_x_N_1.4_O_7.3_Tb. The concentration of the Tb and O elements (in at.%) from the EDS spectra of the Tb_x_O_y_ sample are 29.5 and 79.5, respectively. It is known that in binary Tb oxides, the terbium cations can exist in two states: Tb^3+^ and Tb^4+^ [17]. These various Tb^3+^/Tb^4+^ cationic equilibria lead to different phase transitions, such as TbO_2_, Tb_11_O_20_, Tb_4_O_7,_ and Tb_7_O_12_ [20].

Clusters of small nanoparticles of Tb_4_O_7_ (TEM images) are shown in Figure 7. Terbium oxide prepared from aTbMOF by pyrolysis at 450 °C (sample Tb_x_O_y_) has a fine crystalline structure with crystallite sizes ranging from 5 to 20 nm, as seen in Figure 7a,b. From the results of the selected area, electron diffraction (SAED) (Figure 7c) determined a cubic c-Tb_4_O_7_ structure with a space group of F m-3m (225) [17,43]. The HR TEM image (Figure 7b) showed that the lattice spacings of the nanoparticles were 0.31 and 0.273 nm, close to the values of the interplanar distance of the (111) and (200) planes for the Tb_4_O_7_ nanostructure. Specifically, the average background subtraction filtering (ABSF) image (Figure 7d) indicated an interplanar spacing of 0.308 nm for the (111) plane, 0.271 nm for the (200) plane, and 0.192 nm for the (200) plane of the cubic Tb_4_O_7_. These observations are in agreement with XRD (Figure 2a) and the EDS spectrum in Figure 7e. SEM and TEM results of powders showed the clusters of small Tb_4_O_7_ nanoparticles with various morphologies and irregular shapes with crystallite sizes ranging from 5 to 20 nm in comparison with straw sheaves ~500 nm in length for crystalline cTbMOF and an amorphous morphology of aTbMOF.

### 3.4. Morphological Characterization of Films

SEM surface microstructures of TbMOF (crystalline and amorphous) and terbium oxide films prepared on Pt/SiO_2_/Si are shown in Figure 8. The SEM image of cTbMOF (Figure 8a,b) reveals a straw sheaf-like structure, which is in agreement with that reported in the literature [46]. As shown in Figure 8b, the MOF needle-like particles display different elongated shapes 80–150 nm in size in the porous microstructure of the crystalline film. A SEM image of amorphous MOF film shows that the surface of samples becomes very rough (Figure 8c,d). A similar microstructure with a very rough amorphous MOF surface was observed in aNiFeMOF and aFeCoMOF with worm-like particles [3,6]. SEM image clearly shows the macroporous structure of aMOF, which is formed by worm-like nanoparticles with a diameter of ~100 nm (Figure 8d). In Figure 8e,f we see the macroporous surface microstructures of the amorphous TbMOF film annealed at 450 °C (Tb_x_O_y_-TF sample). A mixture of three terbium oxide structures with different shapes: TbO_x_ (flower-like), Tb_4_O_7_ (flakes), and Tb_2_O_3_ (needles), according to XRD data (Figure 4b), is recorded in the enlarged SEM image (Figure 8f). For comparison, the SEM surfaces of microstructures of the Pt layer on SiO_2_/Si substrates deposited and annealed at 450 °C are observed in Appendix A, respectively. The Pt/SiO_2_/Si substrate secures Pt(111) as nucleation sites and improves the growth of aTbMOF film on its surface (Figure 9a), along with the adhesion between the Tb_x_O_y_ film and Pt layer (Figure 9b). The SiO_2_ layer (~250 nm) is visible below the Pt layer (~20 nm) of the substrate. The thickness of Tb_x_O_y_ and TbMOF films is ~400 nm and ~900 nm, respectively. Appendix A visualizes the size of pores in the range of 73 to 180 nm for the amorphous MOF film. In Appendix A, the particle shape and size of TbO_x_ (flower-like, 119 nm), Tb_2_O_3_ (needles, 183 nm), and Tb_4_O_7_ (flakes, 38 nm) can be observed.

The polycrystalline terbium oxide film differed in composition and consisted of cubic fluorite cF-TbO_x_(111), cF-Tb_2_O_3,_ and c-Tb_4_O_7_ phases [23,24]. The Tb_x_O_y_-TF film grows by a mechanism that is analogous to the Stranski–Krastanov mechanism in that single wetting layer forms of TbO_x_ (x = 1.75), followed by the growth of well-defined, multilayered islands [25,26,47]. Relatively strong interaction with the Pt(111) surface forces the TbO_x_ thin film to adopt a defective fluorite structure rather than the preferred bixbyite (Tb_2_O_3_) structure [47]. Irregularly shaped TbO_x_ islands coexist with smaller flower-like particles [22]. These are crystalline with hexagonal structures concerning the Pt(111) surface, which is consistent with cubic fluorite TbO_x_(111) [22]. The flower-like structure of TbO_x_ consists of plentiful leaf-like nanolayers grown outwardly, which leads to a high surface area. These nanosheets are approximately 30 nm in thickness and 100 nm in width [48]. The formation of the c-Tb_2_O_3_(111) phase (needles) likely occurs through the successive formation of metastable orthorhombic ι-Tb_7_O_12_ and triclinic δ-Tb_11_O_20_ phases, which were consistent with c-Tb_4_O_7_ (flakes) and the variable-density TbO_x_ phase [26].

In Appendix A and Figure 10, the SEM and EDS mapping of the elements in the aTbMOF-TF and Tb_x_O_y_-TF films are shown. EDS spectra of the films show several peaks corresponding to Tb, O, and C elements (Appendix A) and Tb and O (Figure 10), whereas Pt and Si are from the substrate, and the molar ratios of terbium ions matched well with the supposed compound formula. In SEM results, thermal decomposition of worm-like particles in the microstructure of the amorphous aTbMOF-TF film caused the formation of nanoparticles with three terbium oxide structures of different shapes and sizes.

### 3.5. Topography Characterization of Films

The surface topography of the films was investigated by a combination of atomic force (AFM) and confocal microscopy (CM). After the Pt deposition on the SiO_2_/Si substrate, AFM and CM measurements were performed to confirm the coating uniformity and study the homogeneity of the Pt interlayer (Appendix A). AFM revealed the particulate nature of the coating and allowed us to determine the mean size of the particles, which was measured as about 20 nm. One can see the presence of areas with a low surface roughness of the Pt films, giving average roughness of Sa = 1.5 nm (AFM) according to 3 μm × 3 μm and 1 μm × 1 μm scans in Appendix A and Sa = 5.9 nm (CM) by 0.68 mm × 0.48 mm (637 µm × 478 µm) (Appendix A).

Figure 11 shows AFM images of porous TbMOF (crystalline and amorphous) and Tb_x_O_y_-TF thin films. Two-dimensional and three-dimensional AFM images show that the cTbMOF-TF (Figure 11a,b) film surface topography consists of uniform clusters of particles. These areas disappear on the surface of the amorphous film (Figure 11c,d), and its roughness for the same size of scan is higher compared to crystalline MOF film [3,6]. The Pt film on the substrate formed a homogenous nanoparticulate layer, which acts as an intermediate one and improves the adhesion of the amorphous TbMOF-TF film to the substrate. Thermal treatment does not only lead to MOF structure elimination but also decreases the roughness of the Tb_x_O_y_-TF film surface (Figure 11e,f). Sa surface roughness values of the films were calculated as 42.1 nm for cTbMOF-TF, 58.1 nm for aTbMOF-TF, and 53.0 nm for Tb_x_O_y_-TF films from a 3 μm × 3 μm image. According to the roughness of Sa obtained from the CM results shown in Figure 12, the amorphous film (Figure 12a) is very rough and is composed of large clusters, which is contrast to terbium oxide film (Figure 12b).

The combination of the applied spectroscopic methods lets us conclude that we have taken an amorphous TbMOF thin film and subsequently transformed it into oxide film, which was shown to be of rather fine quality. The AFM and CM surface profile corresponds very well with the SEM micrographs of films depicted in Figure 8. Additionally, the SEM and XRD results of films show the formation of TbO_x_, Tb_2_O_3,_ and Tb_4_O_7_ particles in the Tb_x_O_y_-TF film.

Further investigations of samples of thin films can involve luminescence characterization as the terbium mean lifetime and quantum efficiency. TbMOF-TF and terbium oxide films have great application prospects in luminescence sensing, especially in biological and environmental luminescent sensors.

## 4. Conclusions

The crystalline and amorphous terbium metal–organic frameworks (TbMOFs) powders were prepared by solvothermal synthesis using different amounts of sodium acetate (NaOAc) as a modulator. A higher concentration of modulator may lead to the formation of amorphous aTbMOF. A porous amorphous aTbMOF-TF thin film was obtained by deposition from a water suspension of amorphous powder on the SiO_2_/Si substrates using Pt(111) as an interlayer. The amorphous MOF powder and corresponding amorphous film, after pyrolysis at 450 °C in the air, were transformed into terbium oxide Tb_x_O_y_ and Tb_x_O_y_-TF, respectively. The Tb_x_O_y_ powder consists of a single cubic phase c-Tb_4_O_7_ and has a fine crystalline structure with crystallite sizes ranging from 5 to 20 nm, in contrast to the porous Tb_x_O_y_-TF film, which is composed of three different phases, namely, cubic fluorite cF-TbO_x_(111); (x = 1.75), c-Tb_4_O_7_, and c-Tb_2_O_3_. The different morphologies and sizes of particles of flower-like TbO_x_ (~119 nm) islands, needle-like Tb_2_O_3_ (~183 nm), and flake-like Tb_4_O_7_ (~38 nm) were observed. XPS established the presence of Tb in two valence states, Tb^3+^/Tb^4+^, in films.

Thermal treatment not only led to MOF structure transformation but also a reduction in average roughness (Sa) of the Tb_x_O_y_-TF film surface. Sa surface roughness values, obtained by CM, were calculated as 382 nm for aTbMOF-TF and 81 nm for Tb_x_O_y_-TF. The presented methodology is suitable for the preparation of terbium oxide films for optoelectronic and sensor applications.

## Data Availability

The data presented in this study are available on request from the corresponding author. These data are not publicly available due to excessive size and complex format.

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
