# Peer review of "Transformation of Amorphous Terbium Metal–Organic Framework on Terbium Oxide TbOx(111) Thin Film on Pt(111) Substrate: Structure of TbxOy Film"

_nanomaterials, 2022, doi:10.3390/nano12162817_

Round 1

Reviewer 1 Report

Brunckova and co-workers reported the fabrication of TbOx(111) thin film on Pt(111) substrate via transformation of amorphous Tb-MOF. Different phases and structures of TbxOy can be achieved by pyrolysis of amorphous MOF and amorphous MOF thin film. Several issues need to be addressed as following.

 1. How about the pyrolysis transformation of crystalline Tb-MOF instead of amorphous counterparts.

 2. In Figure 2b, the standard index lines of Tb4O7 and fluorite TbOx should be included. Please check the standard index lines of Tb2O3, which fail to match the XRD peaks as indicated by triangle symbols for TbxOy-TF.

 3. In Figure 3, baseline or background line should be properly added before fitting Tb 3d spectra. For TbxOy-TF, the presence of Tb4+ in two different colors is confusing. The deconvolution of Tb 3d should be carefully revised.

4. More analysis including the crystal facets based on the well-defined lattice fringes of TbxOy in the HRTEM image (Figure 7b) should be provided. What is “ABSF” in the figure caption for Figure 7d?

 5. In Figure 8f, how to distinguish between Tb2O3 and Tb4O7 from SEM images?

 6. Please provide some application performance of the as-prepared thin film materials.

Author Response

The Responses to the Reviewers (Manuscript ID: nanomaterials-1868201)

Title: Transformation of Amorphous Terbium Metal-Organic Framework on Terbium Oxide TbOx(111) Thin Film on Pt(111) Substrate: Structure of TbxOy Film

Dear Reviewer,

Thank you for your comments on the manuscript (ID: nanomaterials-1868201) by H. Brunckova et al. I have made some revisions by following your valuable suggestions.

Reviewer 1

Comments and Suggestions for Authors

Brunckova and co-workers reported the fabrication of TbOx(111) thin film on Pt(111) substrate via transformation of amorphous Tb-MOF. Different phases and structures of TbxOy can be achieved by pyrolysis of amorphous MOF and amorphous MOF thin film. Several issues need to be addressed as following.

  1. How about the pyrolysis transformation of crystalline Tb-MOF instead of amorphous counterparts.

A: The crystalline TbMOF has been transformed to terbium oxide by pyrolysis at 450°C in air. Porous bimetallic Ce1-xTbxOy crystals in powder form were prepared by direct pyrolysis of Ce1-xTbx-MOFs precursors [19].

This indicates that after pyrolysis the aTbMOF and cTbMOF precursor have been fully transformed into Tb4O7 with a cubic crystal structure [19,39].

The change is incorporated in the "Results and discussion" section of the Manuscript on page 6 highlighted in green color.

  1. In Figure 2b, the standard index lines of Tb4O7 and fluorite TbOx should be included. Please check the standard index lines of Tb2O3, which fail to match the XRD peaks as indicated by triangle symbols for TbxOy-TF.

A: Thank you for your comment. Figure 2b was corrected. The standard index lines of Tb4O7 and fluorite TbOx (Tb6O11) were included. The triangle symbols assigned Tb2O3, were corrected. Ref. [40] was changed. The intermediate phase Tb6O11 (PDF No 03-065-7172) is assigned as c-fluorite TbOx.

Intermediate Tb11O20, Tb24O44 and Tb48O88 phases in the terbium oxide TbOx system have fluorite structure [40].

[40] Tuenge, R T.; Eyring, L.J. On the structures of the intermediate phases in the terbium oxide system. Solid State Chem. 1982, 41, 75-89. https://doi.org/10.1016/0022-4596(82)90037-8

[48] Lu, X.H.; Li, G.R.; Yu, X.L.; Tong, Y.X. Electrochemical Synthesis and Characterization of TbO2-x Flowerlike Nanostructures. Electrochem. Solid-State Lett. 2008, 11, K85-K88. https://doi.org/10.1149/1.2945878

The change is incorporated in the "Results and discussion" section of the Manuscript on page 6-7 highlighted in green color.

  1. In Figure 3, the baseline or background line should be properly added before fitting Tb 3d spectra. For TbxOy-TF, the presence of Tb4+ in two different colors is confusing. The deconvolution of Tb 3d should be carefully revised.

A: The baseline or background line was removed before fitting Tb 3d spectra for the aTbMOF-TF sample, because the intensity of both peaks (Tb 3d5/2 and Tb 3d3/2) is quite low, and the analysis would be intricate. For TbxOy-TF, the presence of Tb4+ is corrected to one color.

Since the Tb content in the MOF film is low (0.56 at. %), the intensities of both peaks (Tb 3d5/2 and Tb 3d3/2) are low, and the Tb 3d5/2 (1242.5 eV) peak overlaps with the Auger peak of CKLL, therefore the analysis can be realized only over the Tb 3d3/2 peak.

These changes are incorporated in the "Results and discussion" section of the Manuscript on page 7 highlighted in yellow color.

  1. More analysis including the crystal facets based on the well-defined lattice fringes of TbxOy in the HRTEM image (Figure 7b) should be provided. What is “ABSF” in the figure caption for Figure 7d?

A: We include the defined lattice fringes of TbxOy. In order to enhance the signal for noisy low-dose data, average background subtraction filtering (ABSF) was applied to the raw image, and the ABSF-filtered image is shown in Figure 7d.

The HR TEM image (Figure 7b) showed that the lattice spacings of the nanoparticles were 0.31 and 0.273 nm, close to the values of the interplanar distance of the (111) and (200) planes for the Tb4O7 nanostructure. Specifically, the average background subtraction filtering (ABSF) image (Figure 7d) indicated an interplanar spacing of 0.308 nm for the (111) plane, 0.271 nm for the (200) plane, and 0.192 nm for the (200) plane of the cubic Tb4O7.

These changes are incorporated in the "Results and discussion" section of the Manuscript on page 9 highlighted in yellow color.

  1. In Figure 8f, how to distinguish between Tb2O3 and Tb4O7 from SEM images?

A: Based on SEM, TEM, and XRD results, the binary terbium oxide Tb4O7 (Tb2O3.2TbO2) has a fine crystalline structure with crystallite sizes ranging from 5 to 20 nm in TbxOy powder. We assume that in the TbxOy-TF film on Pt(111), the TbOx particles first form, then Tb4O7 (flakes), and then Tb2O3 particles (rods) are formed (Figure 8f).

  1. Please provide some application performance of the as-prepared thin film materials.

A: The luminescence performance of the crystalline mixed (EuGdTb)MOF-TF films we described in previous work [27]. Eu0.5Gd0.25Tb0.25MOF film presented an increase of europium mean lifetime (1.17 ms) by more than 5 times and an improvement for the quantum efficiency (30.3 %) of about 300%, regarding Eu0.25Gd0.5Tb0.25MOF film.

In further investigation of samples of thin films is luminescence characterization as the terbium mean lifetime and quantum efficiency. TbMOF-TF and terbium oxide films have great application prospects in luminescence sensing, especially in biological and environmental luminescent sensors.

These changes are incorporated in the "Results and discussion" section of the Manuscript on page 16 highlighted in yellow color.

Yours sincerely,

Helena Brunckova

Institute of Materials Research

 Slovak Academy of Sciences

Watsonova 47, 040 01 Kosice, Slovakia

Reviewer 2 Report

The manuscript submitted by Brunckova et al. presents the synthesis and structural properties of amorphous terbium metal organic framework thin film (TbMOF-TF) and its transformation to terbium oxide by pyrolysis at 450°C in air. All the synthesized nanomaterials were characterized by fruitful combination of different techniques, including TEM, XPS, XRD, FT-IR, Raman spectroscopy and AFM. But despite to the really quite big amount of data presented there are a list of questions raising upon reading. I would recommend the article presented for publication in the Nanomaterials journal only after the following major issues are taken into account in a revised version.

Common issues:

1.     The introduction part is really complex for the reader. It is not really clear the reasons for the study namely such systems. What should be the application? What is the novelty of the present study? Why the powders and films should be compared? From my opinion the Introduction should include shot description of literature and clearly lays out reasons for studying such systems.

2.     The Abstracts part – the information concerning full set of physical methods used for the samples characterization is absent.

3.     Overall impression – the results discussed in a really thesis. There are mostly just descriptions of the spectra/data obtained. From my opinion each part of the characterization (for different techniques) must contain a clear and concise conclusion at the end concerning the information which it provides.

4. Finally I even did not get to which structure the investigators should move in the future? Which characteristics of the Tb-MOF-TFs are preferred?

More specific:

5.     More information concerning the Pt(111)/SiO2/Si substrate should be included. How the the thickness and structure of Pt layer were confirmed? How is initial SiO2/Si substrate looks like? If it is a plate? What was the thickness of the SiO2 layer? Again concerning the Pt(111) thickness – at line 133 authors claimed that the Pt thickness is about 20 nm. At the same time in XPS spectra there are signals from the Si. Well known that the depth of analysis don't exceed 100-150 Å. How it can be?  

6.     At the Experimental part – Section 2.4. – The information concerning the techniques used for the characterization of the samples is really poor. For example for XPS authors should include which source they used for the measurements? How the quantitative analysis was done? If the authors used the atomic sensitivity factors? Which software used for the peak fittings?

7.     The discussion concerning the XPS data obtained is really complex for understanding. The authors jump from one sample to another as well as from one region to another. The spectra provided in manuscript are fitted, at the same time spectra shown in SI presented without fitting. The BE values in text sometimes don't much the BE values presented in Figures.

8.     I didn't find any information concerning the MOF-Thin film thickness.

Author Response

The Responses to the Reviewers (Manuscript ID: nanomaterials-1868201)

Title: Transformation of Amorphous Terbium Metal-Organic Framework on Terbium Oxide TbOx(111) Thin Film on Pt(111) Substrate: Structure of TbxOy Film

Dear Reviewer,

Thank you for your comments on the manuscript (ID: nanomaterials-1868201) by H. Brunckova et al. I have made some revisions by following your valuable suggestions.

Reviewer 2

Comments and Suggestions for Authors

The manuscript submitted by Brunckova et al. presents the synthesis and structural properties of amorphous terbium metal-organic framework thin film (TbMOF-TF) and its transformation to terbium oxide by pyrolysis at 450°C in air. All the synthesized nanomaterials were characterized by fruitful combination of different techniques, including TEM, XPS, XRD, FT-IR, Raman spectroscopy and AFM. But despite to the really quite big amount of data presented there are a list of questions raising upon reading. I would recommend the article presented for publication in the Nanomaterials journal only after the following major issues are taken into account in a revised version.

Common issues:

  1. The introduction part is really complex for the reader. It is not really clear the reasons for the study namely such systems. What should be the application? What is the novelty of the present study? Why the powders and films should be compared? From my opinion the Introduction should include shot description of literature and clearly lays out reasons for studying such systems.

A: In the Introduction part, we described crystalline and amorphous TbMOFs powders and thin films and their transformation to terbium oxide by pyrolysis. Only a few authors study amorphous aTbMOF-TF thin films.

Metal-organic frameworks (MOFs) have been widely used to prepare corresponding porous metal oxides via thermal treatment. Terbium oxide thin films are potentially attractive materials for the fabrication of multi-layer optical coatings, beam splitters, passive components of integrated circuits, and heat-based laser and recording devices.

This work proposes a method of preparation of aMOFs by the introduction of Tb and offers a strategy through using aTbMOFs instead of cMOFs as a precursor to fabricate crystalline terbium oxide thin films by the pyrolysis process. The TbxOy powders and films were prepared under the same pyrolysis conditions, at 450°C in air, so therefore they were compared. The films are candidates for optoelectronic applications as gas sensors.

We reported a facile strategy to prepare amorphous aMOFs by the introduction of Tb, and further pyrolysis at 450°C to form a unique terbium oxide thin film with a thickness of ~400 nm on Pt/SiO2/Si substrate. The novelty is a fabricated porous film, which is composed of morphologically different phases TbOx, Tb4O7, and Tb2O3 nanoparticles. The prepared films are suitable candidates for optoelectronic applications as gas sensors.

These changes are incorporated in the "Introduction" section of the Manuscript on page 3 highlighted in yellow color.

  1. The Abstracts part – the information concerning the full set of physical methods used for the samples characterization is absent.

A:  The other physical methods used for the samples characterization we added in the revised Abstract. Since the Abstract is supposed to have 200 words, we have removed some parts from the original text of Abstract.

Abstract: This study presents the synthesis and structural properties of amorphous terbium metal-organic framework thin film (TbMOF-TF) and its transformation to terbium oxide by pyrolysis at 450°C in air. The crystalline (cTbMOF) and amorphous (aTbMOF) were prepared by solvothermal synthesis using different amounts (0.4 and 0.7 mmol) of modulator (sodium acetate), respectively. The powders were characterized by differential scanning calorimetry (DSC), thermogravimetry (TG), Fourier transform infrared (FTIR),  Raman spectroscopy and scanning electron microscopy (SEM). Varied chemical composition of surface for TbMOFs and TbxOy was investigated by X-ray photoelectron spectroscopy (XPS). X-ray diffraction (XRD) and transmission electron microscopy (TEM) revealed that aTbMOF has been fully transformed to single Tb4O7 phase with cubic crystal structure at 450°C. Amorphous aTbMOF-TF film was prepared by dropping colloidal solution of amorphous precursor nanocrystals on SiO2/Si substrates with Pt interlayer. XPS confirmed presence of Tb in two Tb+3 and Tb+4 valence states. Amorphous film has rough porous microstructure, which composed of large clusters with worm-like particles, while terbium oxide film consisted of fine crystallites of cubic fluorite cF-TbOx, c-Tb4O7 and c-Tb2O3 phases. The surface topography was investigated by combination of confocal (CM) and atomic force microscopy (AFM). Amorphous film is very rough, opposite crystalline terbium oxide film.

These changes are incorporated in the "Abstract" section of the Manuscript on page 1 highlighted in yellow color.

  1. Overall impression – the results discussed in a really thesis. There are mostly just descriptions of the spectra/data obtained. From my opinion each part of the characterization (for different techniques) must contain a clear and concise conclusion at the end concerning the information which it provides.

A: Thank you for recommendation. We have added a clear and concise conclusion for each part of the characterization (for different techniques).

Based on the DSC and TG results, it was confirmed that the amorphous aTbMOF was thermal decomposed into terbium oxide at 450°C.

Based on the DSC and TG results, it was confirmed that the amorphous aTbMOF was thermally decomposed into terbium oxide at 450°C.

From the FTIR and Raman spectra results, the bands consequent from the presence of ligands in aTbMOF were deleted which is a qualitative confirmation of terbium oxide formation from carbonate structure.

Based on the XRD results, characteristic peaks of terbium oxide were corresponding with the crystalline structure of c-Tb4O7 in the TbxOy precursor, and three phases (cF-TbOx, c-Tb4O7, and c-Tb2O3) in TbxOy-TF film, and was not observed characteristic peaks of aTbMOF.

XPS confirmed the presence of Tb in two Tb+3 and Tb+4 valence states in both films.

SEM and TEM results of powders showed the clusters of small Tb4O7 nanoparticles with various morphology and irregular shape with crystallite sizes ranging from 5 to 20 nm in comparison with straw sheaves in length ~500 nm of crystalline cTbMOF and amorphous morphology of aTbMOF.

As SEM results, thermal decomposition of worm-like particles in the microstructure of amorphous aTbMOF-TF film caused the formation of nanoparticles with three terbium oxide structures of different shapes and sizes.

These changes are incorporated in the "Results and Discussion" section of the Manuscript on page 5, 6, 7, 9, 10, 14, highlighted in yellow color.

  1. Finally I even did not get to which structure the investigators should move in the future? Which characteristics of the Tb-MOF-TFs are preferred?

A: The structure of the terbium oxide film composed of three phases is interesting from the point of view of the presence of Tb3+ and Tb4+ cations. TbMOFs are studied widely due to their outstanding luminescence features, including high-purity color, large Stokes shift, high quantum yields, long decay lifetime, and undisturbed emissive energy. Recently, many TbMOFs were applied as an ideal material for identifying small molecules, metal ions, inorganic anions, organic anions, solvents, gases, and explosives. TbMOF-TFs deposited on substrates of various functions enable different applications. Amorphous and crystalline TbMOF-TF thin films have a different microstructure compared to terbium oxide thin films.

In further investigation of samples of thin films is luminescence characterization as the terbium mean lifetime and quantum efficiency. TbMOF-TF and terbium oxide films have great application prospects in luminescence sensing, especially in biological and environmental luminescent sensors.

These changes are incorporated in the "Results and Discussion" section of the Manuscript on page 16 highlighted in yellow color.

More specific:

  1. More information concerning the Pt(111)/SiO2/Si substrate should be included. How the the thickness and structure of Pt layer were confirmed? How is initial SiO2/Si substrate looks like? If it is a plate? What was the thickness of the SiO2 layer? Again concerning the Pt(111) thickness – at line 133 authors claimed that the Pt thickness is about 20 nm. At the same time in XPS spectra there are signals from the Si. Well known that the depth of analysis don't exceed 100-150 Å. How it can be?

A: We include the information concerning the Pt(111)/SiO2/Si substrate. The thickness and structure of Pt layer were confirmed by SEM, AFM and CM analyses (Figure S8 and Figure S11). The SEM microstructures of Pt/SiO2/Si substrate sample as-deposited and annealed at 450°C was shown in Figure S8. The initial SiO2/Si is plate (p-type silicon [100] single-crystal wafer of diameter 50mm and 270-μm thickness).

In XPS spectra there are signals from the Si due to the fact that the substrate does not have to be evenly covered with film. We indicate the percentage content of elements in at. % for Pt/SiO2/Si substrate (C 27.55%, O 32.4%, Pt 26.96% and Si13.45%).

As the initial substrates, p-type silicon [100] single-crystal wafers of diameter 50 mm and 270 μm thickness were used.

These changes are incorporated in the "Results and Discussion" section of the Manuscript on page 3 highlighted in yellow color.

  1. At the Experimental part – Section 2.4. – The information concerning the techniques used for the characterization of the samples is really poor. For example for XPS authors should include which source they used for the measurements? How the quantitative analysis was done? If the authors used the atomic sensitivity factors? Which software was used for the peak fittings?

A: Thank you for your recommendation. We include the information for XPS analysis. The atomic sensitivity factors were not used.

The additional composition and valence state inquiry was obtained by X-ray photoelectron spectroscopy (XPS). XPS measurements have been carried out on the ESCALAB MkII electron spectrometer, equipped with the twin anode MgKα/AlKα non-monochromated X-ray source. The measurements were performed only with an AlKα X-ray source (1486.6 eV). The energy range was scaled using a standardizing C 1s line of acquired hydrocarbons to 285.0 eV for the electrostatic sample charging. The obtained data were analyzed and processed by SpecsLab2 CasaXPS software (Casa Software Ltd). The processing of the recorded spectra included a subtraction of X-ray satellites and Shirley-type background. The relative concentrations of the different chemical species were determined based on the normalization of the peak areas to their photoionization cross-sections, calculated by Scofield.

These changes are incorporated in the "Materials and Methods" section of the Manuscript on page 4  highlighted in yellow color.

  1. The discussion concerning the XPS data obtained is really complex for understanding. The authors jump from one sample to another as well as from one region to another. The spectra provided in manuscript are fitted, at the same time spectra shown in SI presented without fitting. The BE values in text sometimes don't much the BE values presented in Figures.

A: We corrected text of XPs results. XPS survey spectra of amorphous powder, amorphous film, and terbium oxide film are shown on Figure S3 for basic information. HR XPS spectra of Tb 3d, O 1s, and C 1s (without fitting) for amorphous MOF powder and film and TbxOy-TF film are listed in Figure S4, due to the high of the peaks and their eventual shift of BE energies. The average values of binding energies for Tb 3d, O 1s and C 1s are given in the text.HR XPS spectra of Tb 3d, O 1s and C 1s (fitted) for amorphous and terbium oxide films with the deconvolution analysis of the various element valence states and their content in atomic percentages in surface for booth films.

Figure 3 shows the detailed Tb 3d core-level XPS spectra together with the deconvolution analysis of the various Tb valence states for the booth films. Since the Tb content in the MOF film is low (0.56 at. %), the intensity of both peaks (Tb 3d5/2 and Tb 3d3/2) is quite low, and the Tb 3d5/2 (1242.5 eV) peak overlaps with the Auger peak of CKLL, therefore the analysis can be realized only over the Tb 3d3/2 peak. The Tb 3d5/2 band can be resolved into the two subbands peaking at 1274.6 (Tb3+) and 1276.9 eV (Tb4+). In the terbium oxide film (Tb 9.87 at.%), there exist two principal Tb 3d5/2 and Tb 3d3/2 bands peaking at 1241.8 and 1276.5 eV, respectively. Two subbands peaking at 1238.9 and 1273.7 eV can be ascribed to the Tb3+ state and those two peaking at 1241.8 and 1276.5 eV to the Tb4+ state. According to the XPS analysis results, Zhu et al. [42] reported Tb2O3 (Tb3+) and TbO2 (Tb4+) in the Tb4O7 film.

These changes are incorporated in the "Results and discussion" section of the Manuscript on page 7 highlighted in red color.

  1. I didn't find any information concerning the MOF-Thin film thickness.

A: We include information of  the aTbMOF-TF and TbxOy-TF thin films thickness.

The porous structure of Pt film on SiO2/Si substrate (annealed at 450°C) can provide better adhesion between the TbxOy film and Pt layer (Figure 9b). The SiO2 layer (~250 nm) is visible below the Pt layer (~20 nm) of the substrate. The thickness of TbxOy and TbMOF films is ~400 nm and ~900 nm, respectively.

These changes are incorporated in the "Results and discussion" section of the Manuscript on page 12 highlighted in red color.

Yours sincerely,

Helena Brunckova

Institute of Materials Research

 Slovak Academy of Sciences

Watsonova 47, 040 01 Kosice, Slovakia

Round 2

Reviewer 2 Report

From my opinion current version of manuscript could be accepted for the publication with no further action from authors